# Fracture Load of CAD/CAM Fabricated Cantilever Implant-Supported Zirconia Framework: An In Vitro Study

**DOI:** 10.3390/molecules26082259

**Published:** 2021-04-13

**Authors:** Ibraheem F. Alshiddi, Syed Rashid Habib, Muhammad Sohail Zafar, Salwa Bajunaid, Nawaf Labban, Mohammed Alsarhan

**Affiliations:** 1Department of Prosthetic Dental Sciences, College of Dentistry, King Saud University, Riyadh 11545, Saudi Arabia; ialshiddi@ksu.edu.sa (I.F.A.); sbajunaid@ksu.edu.sa (S.B.); nalabban@ksu.edu.sa (N.L.); 2Department of Restorative Dentistry, College of Dentistry, Taibah University, Al Madinah, Al Munawwarah 41311, Saudi Arabia; drsohail_78@hotmail.com; 3Department of Dental Materials, Islamic International Dental College, Riphah International University, Islamabad 44000, Pakistan; 4Department of Periodontics and Community Dentistry, College of Dentistry, King Saud University, Riyadh 11545, Saudi Arabia; malsarhan@ksu.edu.sa

**Keywords:** zirconium oxide, zirconium, fixed partial denture, dental prosthesis, implant-supported

## Abstract

The fracture resistance of computer-aided designing and computer-aided manufacturing CAD/CAM fabricated implant-supported cantilever zirconia frameworks (ISCZFs) is affected by the size/dimension and the micro cracks produced from diamond burs during the milling process. The present in vitro study investigated the fracture load for different cross-sectional dimensions of connector sites of implant-supported cantilever zirconia frameworks (ISCZFs) with different cantilever lengths (load point). A total of 48 ISCZFs (Cercon, Degudent; Dentsply, Deutschland, Germany) were fabricated by CAD/CAM and divided into four groups based on cantilever length and reinforcement of distal-abutment: Group A: 9 mm cantilever; Group B: 9 mm cantilever with reinforced distal-abutment; Group C: 12 mm cantilever; Group D: 12 mm cantilever with reinforced distal-abutment (*n* = 12). The ISCZFs were loaded using a universal testing machine for recording the fracture load. Descriptive statistics, ANOVA, and Tukey’s test were used for the statistical analysis (*p* < 0.05). Significant variations were found between the fracture loads of the four ISCZFs (*p* = 0.000); Group-C and B were found with the weakest and the strongest distal cantilever frameworks with fracture load of 670.39 ± 130.96 N and 1137.86 ± 127.85 N, respectively. The mean difference of the fracture load between groups A (810.49 + 137.579 N) and B (1137.86 ± 127.85 N) and between C (670.39 ± 130.96 N) and D (914.58 + 149.635 N) was statistically significant (*p* = 0.000). Significant variations in the fracture load between the ISCZFs with different cantilever lengths and thicknesses of the distal abutments were found. Increasing the thickness of the distal abutment only by 0.5 mm reinforces the distal abutments by significantly increasing the fracture load of the ISCZFs. Therefore, an increase in the thickness of the distal abutments is recommended in patients seeking implant-supported distal cantilever fixed prostheses.

## 1. Introduction

There is a substantial increase in the use of zirconia (Zr) dental restorations in the recent years [1] due to the excellent biocompatibility, superior mechanical properties (such as high fracture strength/fracture toughness) and physical properties (dimensional stability, color matching with the teeth and sufficient precision) for dental applications [2]. The use of Zr ceramics is not limited to the single crowns and used for the fabrication of endodontic posts, fixed partial dentures (FPD), implant abutments and frameworks [3,4]. The brittle nature of ceramic leading to the formation of inherent defects and microcracks remain the main obstacle for using the metal-free ceramic restorations [5]. Over time, the microcracks progress, ultimately leading to the fracture and failure of restorations [6]. The Zr dental restorations demonstrated better resistance to crack propagation and inhibition of microcracks by converting the tetragonal-phase to monoclinic-phase (transformation toughening) [7,8,9].

Further advancements in the technology, such as computer-aided designing and computer-aided manufacturing (CAD/CAM), also contributed [10,11,12,13]. The fabrication of an optimal metal ceramic restoration involves a number of complex procedures, which are technique sensitive, time consuming [14] and expensive [15]. With the availability of most advanced CAD/CAM technology, the fabrication of accurately fitting Zr abutments for a long span implant supporting fixed dental prosthesis (FDPs) is feasible [16,17].

Despite excellent mechanical properties, there are various complications associated with the Zr FPDs [18]. Ideally, the FPDs in the posterior region (molars and premolars) should withstand the masticatory forces without mechanical failure [19]. This is critical in terms of the biofuntionality of the posterior restorations that are primarily designed for the functional mastication instead of esthetics [20,21]. The molars are subjected to significantly higher occlusal forces ranging from 300 Newton’s (N) to 800 N compared to the anterior teeth that are subjected to 60 N to 200 N. In fact, the occlusal loads can reach up to 1000 N in certain individuals with parafunctional habits [22,23]. Nevertheless, the data available on the survival rates and complications of Zr FPDs is sparse and controversial, due to the variations in the study designs and variables [24,25].

Cantilever FPDs (CFPDs) are considered as one of the viable treatment options for patients presenting with distal extension edentulous ridges. The reviews regarding the CFPDs are conflicting and some researchers have raised their concerns about the risks associated with CFPDs [26,27]. Cantilevers may adversely influence the biomechanics of implant restorations, leading to mechanical failure or biological complications [28]. The implant-retained prosthesis are particularly beneficial for sites with unfavorable anatomic structures such as patients with excessive ridge bone resorption, proximity of the maxillary-sinus floor, or inferior-alveolar-nerves [29,30]. Nevertheless, the demand for CFPDs has increased due to the benefits including comfort, cost effectiveness and acceptance by the patients [31,32]. The advent of metal free CAD/CAM fabricated, tooth color, implant-supported cantilever zirconia frameworks (ISCZFs), has led to their use in the distal extension free end saddle areas [33]. Currently, evidence is limited regarding ISCZFs with fractures of the framework remaining a major risk involved, with little evidence available on the size and dimensions of the cantilever [34]. Therefore, the purpose of the present in vitro study was to investigate the fracture load for different cross-sectional dimensions of connector sites with different cantilever lengths (load point) of ISCZFs. The null hypothesis was that the different cross-sectional dimensions of the connectors and the cantilever lengths will not affect the fracture load of the ISCZFs.

## 2. Results and Discussion

The weakest ISCZFs were found to be for Group C (10 mm cantilever) with fracture load of 670.39 ± 130.96 N, while the strongest ISCZFs were observed for Group B with a fracture load of 1137.86 ± 127.85 N (Table 1).

The mean fracture load of groups with same cantilever length and reinforced distal abutments showed significant differences (Table 2). The mean differences in the fracture loads between groups A and B (9 mm cantilever) was 326.59 N, which was statistically significant (*p* = 0.000). The comparison between the fracture loads of the Group C and Group D (12 mm cantilever) also showed a statistically significant difference (*p* = 0.000), with a mean difference of 243.95 N. The statistical analysis indicated that the reinforcement of the distal abutments with a greater thickness increases the fracture load of the ISCZFs.

CAD/CAM-fabricated restorations have revolutionized the art and science of construction of indirect restorations. With the CAD/CAM technology, dentists these days can fabricate from simple restorations such as inlays/onlays to a single crown, fixed partial dentures, removal dentures and even maxillofacial prosthesis [10,35]. With the CAD/CAM technology, there are no limitations and the restorations being produced are durable, esthetically pleasing, biocompatible, have better marginal and internal adaptation, and fast fabrication [13,36]. However, the milling procedure involving the cutting of blocks with diamond burs under a torqueing force creates micro cracks that are visually not perceivable, but may become a source of crack propagation and ultimately cause fracture/failure of restoration [37,38].

The present in vitro study investigated the effects of varying cantilever length (load point) and a cross-sectional dimension of connector sites on the fracture load for different ISCZFs. The ultimate fracture strength of the specimens were evaluated using the universal mechanical tester by applying gradual loading until the failure of the specimen. This method has the benefit of being easily manipulated, having a good accuracy and have been previously reported for the assessment of cantilever prosthesis [39,40]. Although the testing method does not fully simulate the complex dynamic masticatory stresses, it can provide an initial assessment of the fracture strength of cantilevered restorations. Due to the brittle nature of ceramics, Zr materials are prone to crack propagation and fracture with a negligible deformation. Therefore, characterization of mechanical properties associated with crack propagation such as fracture strength and fractographic analysis may provide useful information [41].

The results of the present study showed significant variations (*p* = 0.000) in the fracture loads of ISCZFs with a variable cantilever length and connector’s cross-section (Table 2). The Group B showed the highest distal cantilever frameworks fracture loads (1137.86 ± 127.85 N) followed by Group D (914.58 + 149.635 N) and Group A (810.49 + 137.579 N), while Group C specimens showed the weakest fracture load (670.39 ± 130.96 N). These findings suggest that variations in the cantilever lengths and thicknesses significantly affect the fracture load strength of the distal abutments. Therefore, the null hypothesis that different cross-sectional dimensions of the connectors and the cantilever lengths will not affect the fracture load of the ISCZFs, was rejected. In terms of cantilever length, shorter span (7 mm) ISCZFs demonstrated a significant enhancement in the mean fracture load compared to the corresponding longer span (10 mm) ISCZFs, suggesting that reducing the cantilever length of ISCZFs improved the fracture strength of prosthesis. These findings are in agreement with previous studies [42,43] where loading Zr implant frameworks 10 mm from the distal abutment failed at lower loading force than loading at 7 mm from the distal abutment.

Regardless of the ISCZFs’ cantilever length, the reinforcement of the distal abutments (groups B, D) significantly improved the mean fracture load compared to the regular distal abutments (groups A, C). Therefore, the reinforcement of the distal abutments by increasing the connecters’ thickness remarkably improves the rigidity and fracture load of the ISCZFs. Chong et al. [42] compared zirconia implant frameworks of variable connector lengths and dimensions and reported that the thicker (3 × 5 mm) connecters survived a significantly higher fracture load compared to the thinner connecters (3 × 4 mm); however, no association was observed between the length and dimensions of the connecter [42]. These findings validated the results of the present study. Furthermore, a recent study investigating the fracture analysis of the zirconia frameworks also reported the agreeing results [43]. The capability of Zr frameworks to withstand the load to fracture was significantly improved by reducing the cantilever length or increasing the thickness in the occluso-cervical dimension [43]. This evidence suggested that the dimensions of cantilevered restorations may influence the mechanical behavior. Therefore, using the optimal dimensions is vital to increase the restoration longevity and prevent failure due to cyclic masticatory stresses. Considering that the cantilevered prostheses are required to withstand the cyclic masticatory loading for a reasonable period, understanding the association of strength and connector dimension may ensure increased clinical success. Mathematical modeling has been shown to accurately predict the ultimate fracture force of zirconia cantilever implant frameworks under the laboratory test conditions. The physiologic occlusal forces exerted on molar ranges from 810 N to 880 N [44,45]. However, the magnitude of occlusal forces vastly varies among individuals and may also be greater in certain conditions such as clinching or bruxism [46]. Therefore, the enforcement of cantilevered prosthesis is always desired to improve the longevity and clinical success. In the present study, the longer span (10 mm) ISCZFs without reinforcement (group C) were fractured at a force (670.39 N) that is significantly lower than the physiologic masticatory forces of the molar region. On the other hand, the Group B ISCZFs that were reinforced and had a shorter span (7 mm) withstood a significantly greater fracture load than the physiologic masticatory forces exerted in the molar region.

The present in vitro study advocated that the fracture load of the ISCZFs can be remarkably increased by reducing the cantilever span length and increasing the thickness of the distal abutment. Therefore, the dentists, while restoring their missing teeth with distal cantilevers, should consider this important aspect of cantilever design while planning the fabrication of prosthesis.

There are certain limitations of the present study. Although all the specimens were subjected to the artificial aging through storage and thermocycling, the complex nature of the oral cavity and clinical masticatory stresses were not simulated. In the present study, the fracture load was applied gradually at a static rate, while masticatory stresses are cyclic and complex in nature that may influence the outcome in the clinical situation. Testing of the specimens under cyclic loading under the dynamic conditions simulating masticatory stresses would have been valuable and is recommended for future studies. The present study used one standard specimen dimension; however, the ultimate fracture force may by affected due to variations in the pontic designs clinically [13,47]. Similarly, increasing the connector thickness may not be feasible clinically in patients with a limited intra-occlusal space. The current experiment was conducted with the Zr framework only with no ceramic veneering. Furthermore, the abutments were attached to non-original implant analogs, which can be a source of misfit. These limitations may have influenced the final fracture toughness of tested Zr frameworks. To further validate the finding of the present study, additional in vitro and clinical studies simulating the dynamic masticatory forces and an investigation into the associated properties including cyclic fatigue, fracture toughness and flexural strength, are required.

## 3. Materials and Methods

In the present in vitro study, CAD/CAM ISCZF (Cercon, Degudent; Dentsply, Deutschland, Germany) specimens (*n* = 48) were fabricated and divided into four study groups (*n* = 12 each) as follows: Group A (9 mm cantilever length); Group B (9 mm cantilever length with reinforced distal abutment); Group C (12 mm cantilever length); and Group D (12 mm cantilever length, with reinforced distal abutment) (Figure 1). The sample size per group was calculated to be 12 with a total sample size of forty-eight (*N* = 48), using G-power software (G * Power 3.1.9.7, Dusseldorf, Germany) [48] with an effect size of 0.5, power 0.80 and α 0.05.

### 3.1. Sample Preparation

Four wax patterns for ISCZFs representing the study groups were designed and fabricated using modeling wax (GEO Classic opaque wax, Renfert, Hilzingen, Germany). The frameworks were fabricated on two identical cemented-retained titanium abutments (4.2 mm diameter and 2.5 mm height; TiDesigne; Astra Tech EV, Mölndal, Sweden) attached to two implant analogs (4.2 mm and 11 mm length; Astra Tech EV, Mölndal, Sweden). The use of the implant’s analogues for the testing of fracture load and stress distribution for cemented implant-supported crowns is well documented [49]. All the frameworks were designed with predefined dimensions (4 mm height, 3 mm thickness). For the reinforcement of distal abutments (Group B and D), 0.5 mm wax was added all around the distal abutments for the wax framework. The two analogs were mounted parallel to each other with 15 mm apart from their centers, on a rectangular prism-shape (50 × 20 × 20 mm) hard stone (Dento-stone 200; Dentona, Dortmund, Germany). The bottom tip of the two analogues were first mounted on small amount of the hard stone. Then the distance between the two abutments were measured in three different areas using a digital caliper (NB60; Mitutoyo American Corp, Illinois, IL, USA) to confirm that the two analogues are parallel to each other.

For the fabrication of Zr frameworks, the wax frameworks (Figure 2) of all the groups were attached to the scanning frame of a copy-milling machine (Cercon eye, Degudent; Dentsply, Deutschland, Germany). After scanning, the ISCZFs were milled using white Zr blocks (Cercon eye, DeguDent GmbH, Hanau, Germany) and sintered for 6 h at 1350 °C to achieve their full strength. The dimension and length of all the frameworks were confirmed using a digital caliper (NB60; Mitutoyo American Corp, Illinois, IL, USA).

The implant abutments were inserted, secured to the implant analogs with abutment screws (TiDesigne; Astra Tech EV, Mölndal, Sweden), and torqued to 35 Ncm using a manual torque wrench (Astra Tech EV, Mölndal, Sweden). The ISCZFs were then cemented onto the abutments using a hybrid cement (Rely X Unicem, 3M ESPE, Seefeld, Germany) according to the manufacturer’s instructions. The frameworks were seated on the abutments using a standardized vertical load of 5 kg (50 N). The force/load was applied using a customized dental surveyor on the occlusal surface, excess cement was immediately wiped off using a microbrush (Micro-Applicator, 803–165/F, Huanghua Promise Co. Ltd., Hebei, PRC). The specimens were removed once the recommended setting time of 5 min was completed [50].

### 3.2. Thermocycling of the Specimens

Artificial aging of all the specimens was completed as described previously. All the specimens were stored in distilled water for 24 h at 37 °C and thermocycled (Huber, SD Mechatronik Thermocycler, D-83620 Feldkirchen-Westerham, Germany) in water (5 °C and 55 °C) for 6000 cycles with a 30 s dwell time and a 5 s transfer time. The frequency of the thermocycling regime adopted in the current research was based on the assumption that the number of cycles set would represent approximately one year of clinical service for the ceramic specimens [41,42,51].

### 3.3. Testing of the Fracture Strength

A universal testing machine (Instron 5965 testing system, Norwood, MA, USA) was used to test each framework for ultimate fracture force. A tapered-shaped plate was attached to the testing machine and oriented to contact the framework 2 mm away from the end of the specimen (Figure 3). The testing load was a traditional load-to-failure, using a static load that increased incrementally (1 mm/min in a vertical direction to the frameworks) until a fracture occurred [39,40]. The maximum load-to-fracture values were recorded in newtons (N) through computer-generated files attached to the testing machine. Failure was considered by a sharp decrease in force applied to the framework, as measured by the testing machine.

Samples of the failed specimens were analyzed with a scanning electron microscope (SEM) (JEOL, JSM-6360LV, Tokyo, Japan) to determine the mode of the failure (Figure 4). For this purpose, the failed specimens were fixed on a stub, sputter-coated and analyzed by SEM (10 kV, 10X) for surface fractography. The SEM analysis revealed no remarkable differences while comparing the specimens of various study groups.

### 3.4. Statistical Analysis

The fracture load recorded of all the specimens for all the four test groups was statistically analyzed using the SPSS version 22 (IBM., Chicago, IL, USA). The one-way analysis of variance (ANOVA) test was used to determine the statistical differences between the fracture load of the ISCZFs. Multiple comparisons between the fracture loads of all the four test groups was carried out using Post-Hoc Tukey HSD test. The probability for statistical significance was set at α < 0.05.

## 4. Conclusions

Within the limitations of this study, it can be concluded that significant variations in the fracture load between the ISCZFs with different cantilever lengths and thicknesses of the distal abutments were found. Reinforcement of the distal abutments by increasing the thickness of the distal abutment by 0.5 mm significantly increases the fracture load of the ISCZFs. At the same time, increases in the length of the distal cantilever significantly decreases the fracture load of ISCZFs. To ensure the longevity of the ISCZFs, it is recommended to design the distal cantilevers with minimal length. However, when necessary, the fracture load of ISCZFs can be improved by increasing the thickness of the distal abutments in patients seeking for implant-supported distal cantilever-fixed prostheses.

## Figures and Tables

**Figure 1 molecules-26-02259-f001:**
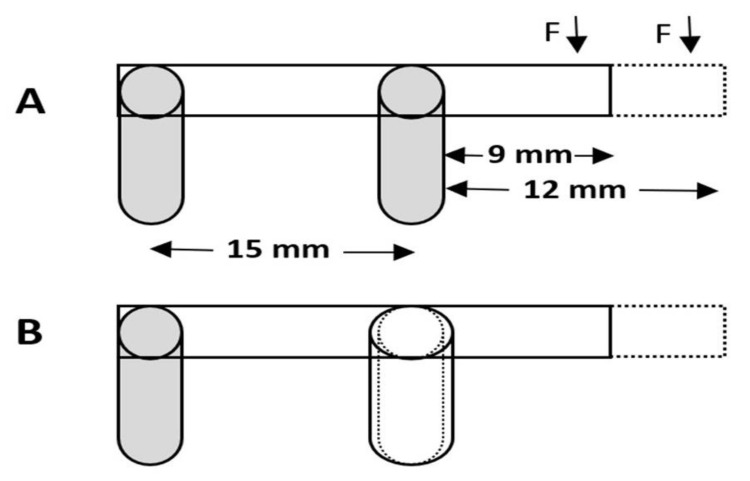
Schematic representation of (**A**): Cantilever frameworks on 2 implant analogs; (**B**): Cantilever frameworks with reinforced distal abutment.

**Figure 2 molecules-26-02259-f002:**
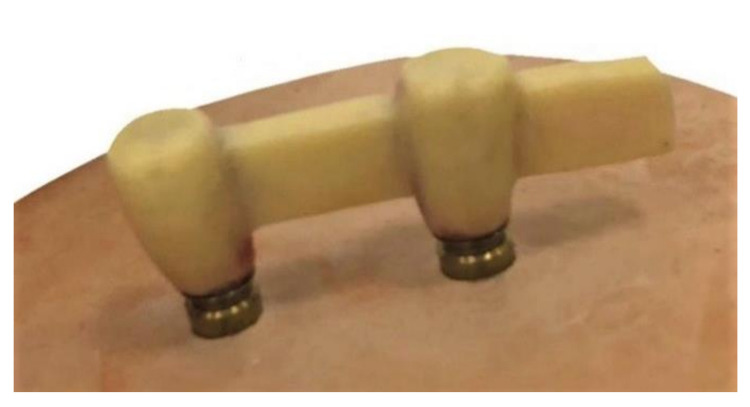
Wax pattern of the cantilever design framework.

**Figure 3 molecules-26-02259-f003:**
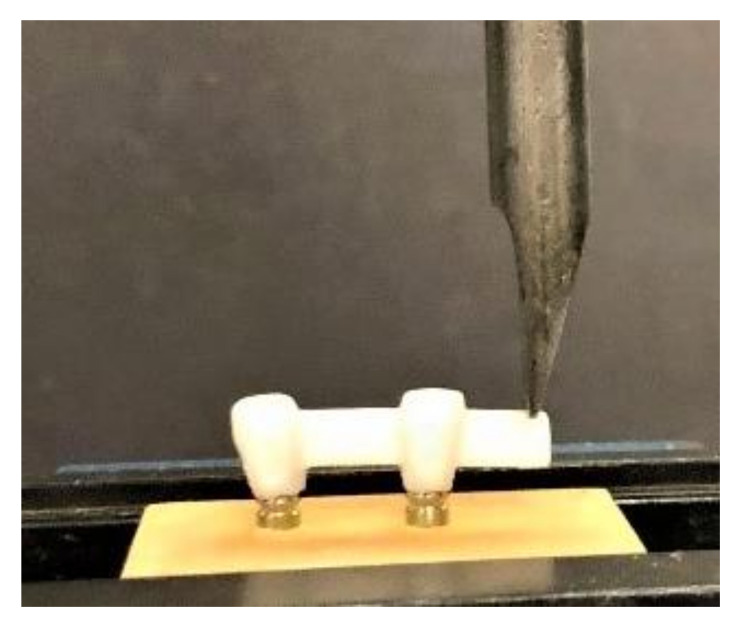
Loading on the specimen using an Instron testing machine.

**Figure 4 molecules-26-02259-f004:**
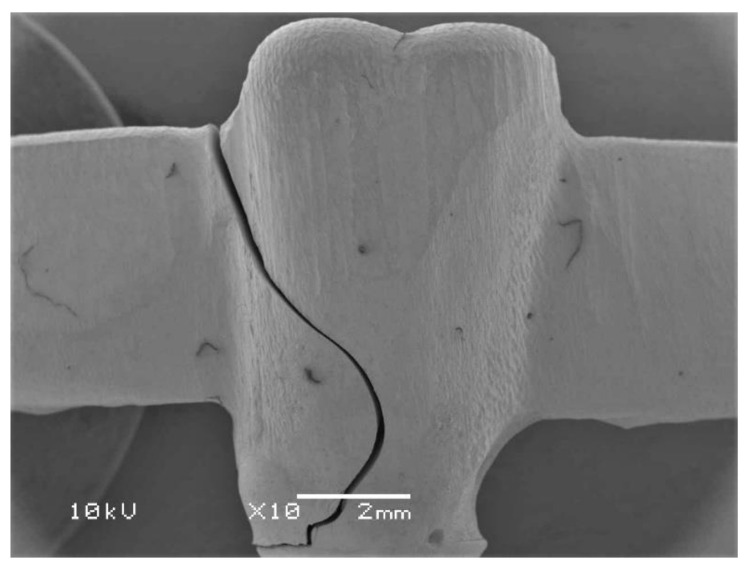
Representative scanning electron microscopic image showing the fractured specimen.

**Table 1 molecules-26-02259-t001:** Descriptive statistics with mean, standard deviation, and ANOVA results of fracture load for the test groups (*N* = 48).

Cantilever Length	Material Groups	*N*	Minimum	Maximum	* Mean	Std. Deviation	95% Confidence Interval for Mean		Anova *p*-Value
							Lower Bound	Upper Bound	
7 mm	Group A	12	540.90	1022.10	810.49	137.579	723.553	898.380	0.000
	Group B	12	950.70	1325.50	1137.86	127.853	1056.326	1218.800	
10 mm	Group C	12	402.80	865.30	670.39	130.963	587.685	754.106	
	Group D	12	609.40	1271.20	914.58	149.635	819.779	1009.927	
	Total	48	402.80	1325.50	883.33	217.084	820.535	946.604	

* Mean fracture load was recorded in Newtons (N).

**Table 2 molecules-26-02259-t002:** Multiple Comparisons and mean differences of the fracture load between the test groups by Post-Hoc Tukey test.

Dependent Variable	Groups	Comparison	Mean Difference	* Significance
Fracture load	Group-A	Group B	−326.59667 *	0.000
		Group C	140.07083	0.072
		Group D	−103.88667	0.260
	Group-B	Group A	326.59667 *	0.000
		Group C	466.66750 *	0.000
		Group D	222.71000 *	0.001
	Group-C	Group A	−140.07083	0.072
		Group B	−466.66750 *	0.000
		Group D	−243.95750 *	0.000
	Group-D	Group A	103.88667	0.260
		Group B	−222.71000 *	0.001
		Group C	243.95750 *	0.000

* The mean difference is significant at the *p* ≤ 0.05 level.

## Data Availability

The data presented in this study are available on the request from the corresponding author.

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
