# Peer review of "Fracture Load of CAD/CAM Fabricated Cantilever Implant-Supported Zirconia Framework: An In Vitro Study"

_molecules, 2021, doi:10.3390/molecules26082259_

Round 1

Reviewer 1 Report

This is an interesting study on the effects of different calntilever lengths on the fracture load-

the study was well executed, however, with several critical issues in the work:

-In the abstract section, an initial sentence on the problem of cad cam and fracture risk must be indicated

-Abstract line 19: do not indicate the numbers of patients, just report them once and, in line 28, indicate the fracture load numerically, much more interesting for readers; finally, in this section, insert a practical clinical reference in the light of the results

- The number of keywords is definitely excessive; reduce them using only Mesh terms from the Pubmed database

-The initial part of the introduction is too repetitive, it needs to be reduced

-Place the hypothesis at the end of the introduction section as a null hypothesis to be refuted in the light of the results obtained from the study

-How was the number of samples chosen?

- Sections 2.1, 2.2 and 2.3 must be inserted in a single section, for example "sample preparations"

-Section 2. “fabricated using ???????? Thing?

-It is not absolutely clear how the CAD-Cam technique was used in the design of the prosthetic reconstructions: improve this aspect

-Section 2.4 even if correctly indicated a previous study report the method in brief

-Indicate a specific section of the SEM analysis by expressly indicating how the samples to be analyzed on the machine are obtained

-The first four lines of the results section should be removed because they are already explained in the introduction

- In the results section, no comment on the results should be made (for example the third sentence)

-In table 1 move the MIn and Max columns before the mean and standard deviation

-Increase the size of the asterisks in table 2

-The part relating to the SEM images is an exercise in style rather than an added factor to the study. In fact, I would have expected first of all the images of the fractures of all the groups. I also ask the authors to devote part of the results and discussion to the possible explanation of any differences. A single image, in fact, as well as a beautiful image, does not add any positive factor to scientific work.

-The discussion section is the one that presents the most problems. After a sentence on the problem that led to the execution of the work, in fact, the results obtained in the study should be immediately described, albeit briefly, focusing on the differences between the different groups

Furthermore, there is no wider discussion of the advantages of CAD-Cam technology in the dental field with its possible applications, both prosthetic and, more generally, clinical. This aspect must be emphasized to increase the attractiveness of scientific work. In this regard, I recommend that you include the following scientific work in the reference section, which may be of help:

Lancellotta V, Pagano S, Tagliaferri L, Piergentini M, Ricci A, Montecchiani S, Saldi S, Chierchini S, Cianetti S, Valentini V, Kovács G, Aristei C. Individual 3-dimensional printed mold for treating hard palate carcinoma with brachytherapy: A clinical report. J Prosthet Dent. 2019 Apr; 121 (4): 690-693. doi: 10.1016 / j.prosdent.2018.06.016. Epub 2018 Nov 30. PMID: 30503148.

-I request a comprehensive review of the English language on all work

Author Response

Response to the reviewers’ comments

Manuscript ID: molecules-1130058

Title: Fracture load of CAD/CAM fabricated cantilever implant-supported zirconia framework: An in vitro study.

Reviewer 1:

This is an interesting study on the effects of different calntilever lengths on the fracture load-

the study was well executed, however, with several critical issues in the work:

Authors’ response: Thank you for reviewing the manuscript and providing insightful comments on our manuscript. The constructive feedback and affirmation of our work has greatly inspired us. Based on the suggestions, authors thoroughly revised the manuscript and provided a point-by-point response to the reviewer’s comments:

 -In the abstract section, an initial sentence on the problem of cad cam and fracture risk must be indicated

Authors’ response: Thank you very much for your remarks. The following sentence is added to the abstract (Line 15-17);

“The fracture resistance of CAD/CAM fabricated implant-supported cantilever zirconia frameworks (ISCZFs) is affected by the size/dimension and the micro cracks produced from diamond burs during the milling process.”

-Abstract line 19: do not indicate the numbers of patients, just report them once and, in line 28, indicate the fracture load numerically, much more interesting for readers; finally, in this section, insert a practical clinical reference in the light of the results

Authors’ response:  As suggested, the mentioned statements are modified; Line 19 is deleted (line 22), required values are added (line 30-31), practical clinical reference in the light of the results is added (line 34-36).

- The number of keywords is definitely excessive; reduce them using only Mesh terms from the Pubmed database

Authors’ response:  The following keywords are added from MeSH terms from the Pubmed database (line 38);

Zirconium Oxide; Zirconium; Fixed Partial Denture; Dental Prosthesis, Implant-Supported.

-The initial part of the introduction is too repetitive, it needs to be reduced

Authors’ response:  Thank you very much for your remarks. the repetitive statements from the introduction are deleted as suggested (Page 2, 2nd paragraph).

-Place the hypothesis at the end of the introduction section as a null hypothesis to be refuted in the light of the results obtained from the study

Authors’ response:  The null hypothesis is added at the end of the introduction section (line 92-94).

-How was the number of samples chosen?

Authors’ response:  Thank you very much for the remarks. The sample size calculation is added (line 101-104).

“The sample size per group was calculated to be 12 with a total sample size of forty-eight (N=48), using G-power software (G*Power 3.1.9.7, Germany) with effect size of 0.5, power 0.80 and α 0.05.”

- Sections 2.1, 2.2 and 2.3 must be inserted in a single section, for example "sample preparations"

Authors’ response:  As suggested, sections 2.1, 2.2 and 2.3 are merged as a single section 2.1 Sample preparation (line 125).

-Section 2. “fabricated using ???????? Thing?

Authors’ response:  The missing information has been added to the mentioned statement (line 127).

-It is not absolutely clear how the CAD-Cam technique was used in the design of the prosthetic reconstructions: improve this aspect

Authors’ response:  Thank you very much for your remarks. After scanning the wax patterns the frameworks were milled. This information is now added to methodology (line 135-141).

-Section 2.4 even if correctly indicated a previous study report the method in brief

Authors’ response:  Thank you very much for your remarks. We have revised the relevant section and added more details about the method (section 2.2; line 175-76).

-Indicate a specific section of the SEM analysis by expressly indicating how the samples to be analyzed on the machine are obtained

Authors’ response: Thank you very much for the insightful suggestion, author have included further details the SEM analysis (line 192-93).

-The first four lines of the results section should be removed because they are already explained in the introduction

Authors’ response:  As suggested, the first four lines of the results section are removed (line 237-40).

- In the results section, no comment on the results should be made (for example the third sentence)

Authors’ response:  Authors fully agree with the reviewer’s point and accordingly deleted the mentioned statement (line 241).

-In table 1 move the MIn and Max columns before the mean and standard deviation

Authors’ response:  Thank you very much for your remarks. The corrections in table are incorporated (Table 1).

-Increase the size of the asterisks in table 2

Authors’ response:  The mentioned change has been made (Table 2).

-The part relating to the SEM images is an exercise in style rather than an added factor to the study. In fact, I would have expected first of all the images of the fractures of all the groups. I also ask the authors to devote part of the results and discussion to the possible explanation of any differences. A single image, in fact, as well as a beautiful image, does not add any positive factor to scientific work.

Authors’ response:  Authors are very much thankful for the valuable comments regarding the presentation of SEM data. Authors fully agree with the reviewer’s point that any differences between all the groups’ specimens should be discussed in context. Authors collected the SEM data to compare the morphological features of fractured surfaces of all the failed specimens. However, the SEM analysis revealed no remarkable differences while comparing various groups (line 193). All the specimens showed a similar pattern of fracture (oblique fracture) from occlusal to cervical direction. Considering there were no significant differences in the fracture pattern of various groups, authors only presented a representative SEM image to assist the readers with the pattern of fracture in such type of fracture loading.

-The discussion section is the one that presents the most problems. After a sentence on the problem that led to the execution of the work, in fact, the results obtained in the study should be immediately described, albeit briefly, focusing on the differences between the different groups

Authors’ response:  Authors are thankful for the insightful feedback, as suggested, the results obtained in the study are added briefly after the mentioned statement (line 287-290).

Furthermore, there is no wider discussion of the advantages of CAD-Cam technology in the dental field with its possible applications, both prosthetic and, more generally, clinical. This aspect must be emphasized to increase the attractiveness of scientific work. In this regard, I recommend that you include the following scientific work in the reference section, which may be of help:

Lancellotta V, Pagano S, Tagliaferri L, Piergentini M, Ricci A, Montecchiani S, Saldi S, Chierchini S, Cianetti S, Valentini V, Kovács G, Aristei C. Individual 3-dimensional printed mold for treating hard palate carcinoma with brachytherapy: A clinical report. J Prosthet Dent. 2019 Apr; 121 (4): 690-693. doi: 10.1016 / j.prosdent.2018.06.016. Epub 2018 Nov 30. PMID: 30503148.

Authors’ response: Thank you very much for the valuable comments and providing the valuable reference. Accordingly, authors have added the advantages of CAD/CAM technology in the dental field with its prosthetic and, more generally, clinical applications (page 7-8). In addition, the suggested study (Ref#10 of the revised file) and some other updated references have been cited (Ref# 10, 13, 43, 44, 45, and 46).

-I request a comprehensive review of the English language on all work

Authors’ response:  The revised manuscript is proofread by a native speaker of English language and all the grammatical errors and typos have been corrected throughout the manuscript (all corrections made are in visible in track changes).

Finally, the authors would like to thank the reviewer for the constructive feedback and helping the authors to improve the contents and quality of this manuscript. We hope the quality of the manuscript has been improved and will be acceptable for publication.

Sincerely,

Prof. S R Habib

Reviewer 2 Report

I find the article interesting but it must be improved for acceptance in the journal

in the abstract remove the n = 12 from the place where it is now and put it when describing the groups.
try not to put abbreviations in the abstract
In line 42 of the introduction cite the article of

Clinical outcomes of veneered zirconia anterior partial fixed dental prostheses: A 12-year prospective clinical trial
María Fernanda Solá-Ruiz et al. J Prosthet Dent. 2021.
Prospective study of monolithic zirconia crowns: clinical behavior and survival rate at a 5-year follow-up Mª Fernanda Solá-Ruiz et al. J Prosthodont Res. 2020.   in line 49 cite the article of Zirconia in fixed prosthesis. A literature review Rubén Agustín-Panadero et al. J Clin Exp Dent. 2014.
in material and methods explain how the implant analogs were mounted in the specimens and describe well the fixation material of the analogs in the sample.
why it was used analogues and not original implants. It is proven that it can affect the results of in vitro research. have they been based on the ISO standard?
in the static load because they use a crosshead speed of 1mm / min and not 0.5mm / min. I think that a high speed can influence the final results of the investigation.
This study has the limitation of not having subjected the specimens to dynamic fatigue load. in discussion show the limitations of your study and in vitro investigations of the style of your study. also, re-mark the clinical implication of the results obtained. the acceptance or rejection of the proposed work hypothesis must be made clear in the discussion section. In the conclusions section, it is important to specify the importance, not only of the reinforcement of the distal abutment, but also of the influence of the cantilever extension.

Author Response

Response to the reviewers’ comments

Manuscript ID: molecules-1130058

Title: Fracture load of CAD/CAM fabricated cantilever implant-supported zirconia framework: An in vitro study.

Reviewer 2:

I find the article interesting but it must be improved for acceptance in the journal

Authors’ response: Thank you for reviewing the manuscript and providing insightful comments on our manuscript. The constructive feedback and affirmation of our work has greatly inspired us. Based on the suggestions, authors thoroughly revised the manuscript and provided a point-by-point response to the reviewer’s comments:

In the abstract remove the n = 12 from the place where it is now and put it when describing the groups. try not to put abbreviations in the abstract

Authors’ response:  Thank you very much for your remarks. The n=12 has been shifted. Regarding the abbreviations as the two words [computer aided designing and computer aided manufacturing (CAD/CAM); implant-supported cantilever zirconia frameworks (ISCZFs)] are too lengthy, they have been abbreviated first then their abbreviations are used in the abstract. It is for the convenience of the reader. Thank you for understanding.

In line 42 of the introduction cite the article of

  • Clinical outcomes of veneered zirconia anterior partial fixed dental prostheses: A 12-year prospective clinical trial María Fernanda Solá-Ruiz et al. J Prosthet Dent. 2021.
  • Prospective study of monolithic zirconia crowns: clinical behavior and survival rate at a 5-year follow-up Mª Fernanda Solá-Ruiz et al. J Prosthodont Res. 2020.

Authors’ response:  Thank you very much for the insightful suggestion, the mentioned references are added (line 48; Ref # 3,4).

in line 49 cite the article of Zirconia in fixed prosthesis. A literature review Rubén Agustín-Panadero et al. J Clin Exp Dent. 2014.

Authors’ response:  Thank you very much for the insightful suggestion, the mentioned references are added (line 54; ref# 7).

in material and methods explain how the implant analogs were mounted in the specimens and describe well the fixation material of the analogs in the sample.

Authors’ response:  Thank you very much for your remarks. The following paragraph is added to the section 2.1 of the methodology section (line 135-40).

“The two analogs were mounted parallel to each other with 15 mm apart from their centers, on a rectangular prism-shape (50×20×20 mm) hard stone (Dento-stone 200; Dentona, Dortmund, Germany). The bottom tip of the two analogues first mounted on small amount of the hard stone. Then the distance between the two abutments were measured in three different area using a digital caliper (NB60; Mitutoyo American Corp, USA) to confirm that the two analogues are parallel to each other.”

why it was used analogues and not original implants. It is proven that it can affect the results of in vitro research. have they been based on the ISO standard?

Authors’ response:  Thank you very much for the remarks. The use of implant's analogues or abutment's analogues does not influence the fracture load and stress distribution for cemented implant-supported crowns as reported previously (Rafael et al 2017). This point is added to methodology with the following reference (line 132, ref#36).

in the static load because they use a crosshead speed of 1mm / min and not 0.5mm / min. I think that a high speed can influence the final results of the investigation.

Authors’ response:  Thank you very much for your remarks. The authors agree with the reviewer on this point. However, in this study the setting of speed was based on the speed settings of 1-2mm/min reported in some previous research studies (ref# 42). In addition, the influence of crosshead speed on the fracture load can be investigated in future studies.

This study has the limitation of not having subjected the specimens to dynamic fatigue load. in discussion show the limitations of your study and in vitro investigations of the style of your study. also, re-mark the clinical implication of the results obtained. the acceptance or rejection of the proposed work hypothesis must be made clear in the discussion section.

Authors’ response:  Thank you very much for the valuable comments about the study limitations. Accordingly, the null hypothesis and following points are added to discussion and limitations/recommendations.

“Therefore, the dentists while restoring their missing teeth with distal cantilevers should consider this important aspect of cantilever design while planning the fabrication of prosthesis” (line 333-335).

“Testing of the specimens under cyclic loading would have been valuable and is recommended for future studies” (line 340-343).

In the conclusions section, it is important to specify the importance, not only of the reinforcement of the distal abutment, but also of the influence of the cantilever extension.

Authors’ response:  As suggested, the information about the influence of the cantilever extension has been added to the conclusions section.

Finally, the authors would like to thank the reviewer for the constructive feedback and helping the authors to improve the contents and quality of this manuscript. We hope the quality of the manuscript has been improved and will be acceptable for publication.

Sincerely,

Prof. S R Habib

Round 2

Reviewer 1 Report

i reccommend work acceptation

Reviewer 2 Report

the manuscript is suitable for publication